# Feasibility of Brachial Occlusion Technique for Beat-to-Beat Pulse Wave Analysis

**DOI:** 10.3390/s22197285

**Published:** 2022-09-26

**Authors:** Lukas Matera, Pavol Sajgalik, Vratislav Fabian, Yegor Mikhailov, David Zemanek, Bruce D. Johnson

**Affiliations:** 1Department of Physics, Faculty of Electrical Engineering, Czech Technical University in Prague, 16627 Prague, Czech Republic; 2Department of Cardiovascular Diseases, Mayo Clinic, Rochester, MN 55905, USA; 32nd Department of Internal Medicine—Cardiology and Angiology of General University Hospital and 1st Medical Faculty of Charles University, 12808 Prague, Czech Republic

**Keywords:** occlusion, cuff, non-invasive, pulse wave

## Abstract

Czech physiologist Penaz tried to overcome limitations of invasive pulse-contour methods (PCM) in clinical applications by a non-invasive method (finger mounted BP cuff) for continuous arterial waveform detection and beat-to-beat analysis. This discovery resulted in significant interest in human physiology and non-invasive examination of hemodynamic parameters, however has limitations because of the distal BP recording using a volume-clamp method. Thus, we propose a validation of beat-to-beat signal analysis acquired by novel a brachial occlusion-cuff (suprasystolic) principle and signal obtained from Finapres during a forced expiratory effort against an obstructed airway (Valsalva maneuver). Twelve healthy adult subjects [2 females, age = (27.2 ± 5.1) years] were in the upright siting position, breathe through the mouthpiece (simultaneously acquisition by brachial blood pressure monitor and Finapres) and at a defined time were asked to generate positive mouth pressure for 20 s (Valsalva). For the purpose of signal analysis, we proposed parameter a “Occlusion Cuff Index” (OCCI). The assumption about similarities between measured signals (suprasystolic brachial pulse waves amplitudes and Finapres’s MAP) were proved by averaged Pearson’s correlation coefficient ($\overset{-}{r}$ = 0.60, *p* < 0.001). The averaged Pearson’s correlation coefficient for the comparative analysis of OCCI between methods was $\overset{-}{r}$ = 0.88, *p* < 0.001. The average percent change of OCCI during maneuver: 8% increase, 19% decrease and percent change of max/min ratio is 35%. The investigation of brachial pulse waves measured by novel brachial blood pressure monitor shows positive correlation with Finapres and the parameter OCCI shows promise as an index, which could describe changes during beat-to-beat cardiac cycles.

## 1. Introduction

In current clinical practice mathematical algorithms for hemodynamic assessment from the intra-arterial blood pressure waveform has been successfully implemented in devices used predominantly in the intensive care units [1,2,3,4]. This waveform is a result of interaction between stroke volume and arterial compliance, characteristic impedance, and systemic vascular resistance. These devices use this waveform for estimating important hemodynamic parameters (SV, SVV, PPV, SVR, MAP, CO, etc.). Some of them needs special equipment for work, such their own catheters with thermistors on their tips, but others use their own transducers, which are attached to ordinarily catheters used during femoral or radial catheterization. Most of these devices need initial calibrations by thermodilution or transpulmonary lithium indicator dilution method. The most common are devices such as FloTrac/Vigileo (Edwards Lifesciences, Irvine, CA, USA), PiCCO2 (Pulsion Medical Systems AG, Feldkirchen, Germany), LiDCOplus PulseCO (LiDCO Ltd., London, UK) or MostCare^UP^ (Vytech, Padova, Italy) [5,6,7,8]. However, the invasive nature of signal acquisition represents a drawback for a broader practical use of pulse contour methods (PCM) in hospital environment (cardiology, nephrology, etc.). To overcome limitations of invasive PCM methods, such as need of the art-line and initial calibration, a non-invasive method for continuous arterial waveform detection based on volume clamp principle, first described by Czech physiologist Penaz, has been introduced and demonstrated feasibility for clinical applications [5,6]. This technique is based on combination of finger PPG device and servo volume-clamp mechanism with inflatable finger cuff to determine the pulse pressure waveform. The PPG device detects changes during the cardiac cycle in arterial volume and the servo mechanism immediately reacts by increase/decrease of the pressure in finger cuff to prevent change in its volume. Modelflow method or other algorithms are than used to determine hemodynamic parameters [5,6,7]. Nevertheless, this method relies significantly on quality of finger BP recording, which can be limited due to any cause of profound centralization of circulation. Restricted peripheral circulation in cold conditions and finger movements during exercise are known sources of error readings [9,10]. Although this technique has excellent use for beat-to-beat stroke volume tracking, relatively expensive costs and operator’s skills requirements reserved this approach predominantly for research purposes. A stroke volume variation (SVV) and pulse pressure variation (PPV) has gained significant interest in early era of human physiology discoveries to be useful in distinguishing between healthy and diseased heart conditions [11,12,13,14]. In particular, a maneuver, consisting of sustained forced expiratory effort against an obstructed airway, called Valsalva, has been shown to carry remarkable diagnostic value in cardiology due to thoughtfully described arterial blood pressure response in patients with altered hemodynamics (for instance congestive heart failure, pulmonary hypertension, aortic stenosis, etc.) [15,16,17]. Simple and affordable techniques are currently missing on the market to allow application of body of research around SVV for diagnostic purposes in a wider population. The brachial occlusion method, originally developed for a non-invasive assessment of augmentation index and central blood pressure, yields a clean arterial pressure waveform well correlating to the invasively measured arterial curve [18]. It has been also demonstrated that this technique has potential for non-invasive assessment of cardiac output at rest and during exercise and pulse wave velocity (PWV) assessment and due to a more centrally located signal acquisition this may provide certain advantages over the finger cuff techniques [19,20,21]. We hypothesize that the brachial occlusion cuff technique may demonstrate similar results in PPV analysis compared to previously used volume-clam finger cuff method. This could allow for less cumbersome setup without requiring servo-regulated system and may ultimately outperform volume-clamp method in signal acquisition during relative acral hypothermia or ischemia. Thus, the aim of this work is to validate signals obtained via proposed technique and the established technique using the Finapres in respect to the beat-to-beat pulse wave analysis during voluntary breathing perturbations.

## 2. Materials and Methods

### 2.1. Subjects

A total of twelve healthy adult subjects (2 females, age = (27.2 ± 5.1) [years]; BMI = (25.8 ± 2.5) [kg·m^−2^]) with no history of tobacco use were recruited from the surrounding community of Rochester, MN. There were no exclusion criteria. All aspects of the study were approved by the Mayo Clinic Institutional Review Board and conform to the Helsinki declaration. After the measurements, the subjects whose data was not able to evaluate because of erroneous readings either from experimental device or Finapres were discarded. In this study it was one subject, and it was because of underinflated brachial arm cuff.

### 2.2. Experimental Devices

The non-invasive blood pressure waveform was obtained by our developed experimental device using the suprasystolic occlusion principle as previously described [18,19,20]. This suprasystolic pressure completely occlude the brachial artery and propagating pulse waves, as a result of interaction of stroke volume with cardiovascular system, are completely transmitted through the pressurized arm cuff to our differential pressure sensor. The key aspect of our method—differential pressure sensor with a range of a few mm Hg—than compare input from brachial arm cuff with suprasystolic pressure and on its output is the sensitive brachial suprasystolic pulse wave. These pulse waves reflect the state of the systemic arteries, where the waves are repeatedly reflected at branching or pathologies (stenosis, aneurysms, atherosclerotic plaques, etc.) in their arterial walls. In this way, the experimental blood pressure monitor is able to detect even very weak pressure pulsations and based on our previous studies, we proved that this method can be used for determination of a few hemodynamics parameters in time or frequential domain and especially for PWV assessment [19,20,21,22], see Figure 1.

An appropriately sized inflatable pneumatic cuff (CM2, Omron Healthcare Group, Kyoto, Japan) was positioned on the upper arm such that a 5 cm overlap of the cuff was present at 3 cm above the cubital fossa. The brachial cuff was interfaced with the experimental blood pressure monitor [19,20]. The final raw pressure signal went through an analog low-pass noise filter with a cut-off frequency of 650 Hz. On the contralateral upper extremity was attached appropriately sized finger cuff of the Finapres system (Finapres Medical Systems B.V., Enschede, The Netherlands) [6]. In addition, mouth pressure was monitored via mouthpiece equipped with pressure/flow sensors (MGC Diagnostics, St. Paul, MN, USA). For the simultaneous recording of all signals digitized at a sampling frequency of 1000 Hz, the PowerLab 16/30 (AD Instruments, Bella Vista, NSW, Australia) was employed.

### 2.3. Study Design

The present investigation was a single-center, prospective non-randomized study. Data were collected in the upright siting position. After 10–15 min of quiet rest (baseline) subjects were asked to breathe through the mouthpiece and the brachial cuff was inflated to suprasystolic pressure for duration of ~50 s after prior manual BP assessment. During this period, the Finapres signal was simultaneously acquired to record SVV during spontaneous breathing. Secondly, after prior training, subjects were asked to generate positive mouth pressure against the mouthpiece occlusion of ~20 mmHg (Valsalva maneuver). Positive mouth pressure was held for 20 s once the brachial cuff was inflated and stabilized at the suprasystolic pressure. Mouth pressure was controlled by subjects using unblinded visualization of the mouth pressure signal. Both modes of tests (at rest and during Valsalva) were recorded in triplicate with 5 min periods of rest with the brachial cuff deflated between recordings allowing normalization of blood circulation during the arm. Patients were encouraged to remain still, avoid coughing, and relax their arm to minimize motion artefacts during the recordings.

### 2.4. Signal Processing

The program MATLAB (The MathWorks, Inc., Natick, MA, USA) was used to provide signal analysis of measured data described in the study design. For these purposes were created algorithms for data pre-processing and analysis in the time and frequency domain in order to apply same parameters for data measured by novel brachial cuff method and data determined by Finapres. Algorithms were part of our solution, which was the MATLAB GUI application for analysis of suprasystolic pressure pulsations and their comparison with datasets from other invasive/noninvasive techniques for hemodynamic parameters assessment. The channels must be synchronized with the same device. In our case it was already mentioned PowerLab 16/30. Algorithms use regularly known facts in the signal processing and analysis, and we implemented them in our application, as described follows. According to the spectral analysis of the signals and HR were counted the cut-off frequencies for the removing of the baseline wander (breathing, random body motion and general muscle contraction) and high frequency electrical noise. These cut-off frequencies were the same for each patient data analysis. The pulsation detection algorithms were applied for precise flattening of every suprasystolic pulsation. The desired suprasystolic signal (pulsations captured by differential pressure sensor between closing and opening ‘closing valve’ [23]) after equalization of pressures was used for data analysis. The potential peaks were limited by the ‘duration’ condition in the time domain: the possible interval of HR was set to (30–200) [beats/min]. Other conditions were implemented in the data analysis, where could be parameters, determined from the suprasystolic pulse waves, out of bounds. For the suprasystolic pulse waves were found their matching Finapres pulse waves in the simultaneously recorded signals. If there were any signal abnormalities in pulse waves captured by either of the devices, both pulse waves were discarded from comparison, see Figure 2. 

According to study [19] was defined ratio-systolic area under the individual pulse wave divided by its amplitude. This ratio was defined as an Occlusion Cuff Index (OCCI) and was counted for both methods, see Figure 3. In the Finapres signal it means ratio SV/PP. To get only the systolic part of pulse waves, the dicrotic notch detection algorithms were used, and the signals were interpolated after the dicrotic notch according to normal distribution decrease gradient.

For the insight into the changes during maneuvers, it was used three-point moving average of OCCI during maneuver and calculated percent change of its maximum and minimum. To comparison with the OCCI in steady state also three-point moving average before maneuver was used. 

### 2.5. Statistical Analysis

In order to achieve the purpose of the study, we assumed hypothesis that parameters obtained from brachial-cuff based measurement correlates with parameters obtained by Finapres. We performed the Pearson’s correlation analysis to confirm the assumption. Statistically significant results were those with *p* < 0.05. The main studied parameters for comparison were amplitudes of pulse waves, MAP and OCCI. The MAP of the Finapres was counted as an integral from the pulsation divided by its duration. Parameters, such as RCT, amplitude or specified higher differences between following pulse waves, were used to describe changes aroused by Valsalva maneuver. In addition, these parameters were also used for elimination of pulse waves, which had outliers. 

Descriptive statistics was used to determine the max, min, SD, etc., of specified parameters and also for the description of study population. The upper and lower limits of agreement in parameters were ±1.96 SD.

To calculate Pearson’s correlation coefficient between methods for the whole measurements, we performed Fisher’s Z-transformation. This transformation is provided in calculation of average of Pearson’s correlation coefficient values for the n-independent measurements [24]. The complete statistical analysis performs MATLAB Inc. software with its various toolboxes. 

## 3. Results

To describe the results in the graphs, a few determined parameters were normalized. In the case of parameters (amplitude of the suprasystolic pulsations, MAP of the same wave measured by Finapres, see Figure 2) and after the use of the Fisher’s Z-transformation, the averaged Pearson’s correlation coefficient parameters was $\overset{-}{r}$ = 0.6, *p* < 0.001, see Table 1. This result confirms the hypothesis about similarities between measured pulse waves by different methods. 

The key goal of the study—the percent changes of suprasystolic OCCI during Valsalva in the healthy subjects, see Table 2. The OCCI increases during maneuver average of 8%, the decrease is there 19% and the ratio between max and min during maneuver is 35%, see Figure 3. The position of three max OCCI pulse waves are most often in the start of the maneuver and the three mins occurs in the end. The averaged Pearson’s correlation coefficient for the comparative analysis of OCCI between methods was $\overset{-}{r}$ = 0.88, *p* < 0.001, see Figure 4. The number of peaks included in the correlation through all measurements were 1737.

## 4. Discussion

The aim of this study was to validate the non-invasive continuous blood pressure signal obtained via brachial cuff occlusion method with a signal acquired from reference device, Finapres (Finapres Medical Systems B.V., Enschede, The Netherlands), during voluntary changes of intrathoracic pressure caused by Valsalva maneuver [21]. Finapres is a validated device with widely use in clinical practice for beat-to-beat analysis [25,26]. We can confirm hypothesis about suprasystolic brachial pressure pulsations and their correlation with pulsations from Finapres, $\overset{-}{r}$ = 0.6 (*p* < 0.001). The focus was on the amplitude of the suprasystolic brachial pressure pulsations and Finapres’s MAP, as an important indicator of cardiovascular health, an indicator of perfusion of the organs and parameter for the determination of other hemodynamic parameters. Comparison was provided by Pearson’s correlation coefficient and its Fisher Z-transformation for averaging correlation coefficient, based on the method, which describes minimum-variance unbiased solution [24]. It confirms that suprasystolic pulsations can give us some important information about hemodynamic parameters, which can be obtained by Finapres measurement. Figure 1 shows the example of suprasystolic pulsation with normalized amplitude. Even though you cannot determine the blood pressure from these pulsations as you can from Finapres pulsations, the importance is in theirs shapes, which describe very sensitively their behavior in the arterial tree (propagation and reflection). Creating conditions, which non-invasively give some load to the cardiovascular system, such as breathing perturbation (Valsalva maneuver affects heart rate and blood pressure), can lead to important findings about cardiovascular system reaction. Because of the interest in the changes (percent changes) of the system during this load, the focus is on the ratios of specified studied parameters.

The OCCI ratio, previously described [19], is sensitive to the morphology of systolic part of the arterial pressure waves. In the Finapres signal this parameter is represented by the SV/PP. This ratio or inverse value (arterial stiffness) of this ratio is mentioned in the studies about cardiovascular risks and hypertension in echocardiographic examinations [27,28]. Pulse pressure (PP) is in these methods based on the peripheral BP measurements, so the accuracy of these estimations are dependent on the amplifications between aorta and peripheral arteries. Stroke volume (SV) in this formula is mostly calculated by invasively methods in the aorta. It is therefore difficult to determine SV/PP.

In the novel method, the suprasystolic brachial occlusion causes, that the measured pulsations are closely related to the aortic pulsations, which differ from each other by transfer function based on the pulse waves decomposition to propagating and reflecting wave. The suprasystolic pressure and differential pressure sensor are crucial to detect these reflected waves, which main reflection is supposed to be from iliac bifurcation [21,29]. For the method comparison was used the whole systolic part of suprasystolic pulsations. This closer distance to aortic pulsations and closer relation leads to less influenced amplitude of the pulse wave by amplification in peripheral site and more accurate ratio OCCI.

Across the measurement in study population, we confirmed by novel method the general facts about behavior of hemodynamics during Valsalva maneuver. There is a change in amplitude of the suprasystolic pulse waves and change of the HR. This relation is described in many studies [25,30], which took care about hemodynamics during Valsalva maneuver. The response to the maneuver could be divided into four stages, in which take turns the rises of HR and BP with their decreases. During these hemodynamic changes, we get into the limitations of both techniques. 

Diastolic pressure decay differences in various parts of arterial tree have been previous described due to factors substantially affecting diastolic pressure decay properties locally. Therefore, this represents certain physiologic limitation to a high correlation between signal from brachial and digital arteries in beat-to-beat analysis [31].

### Limitations

In the complex methodology, which was composed for this study, was the major obstacle to guarantee the suprasystolic cuff pressure during all simultaneous operations. In one case, there was only systolic pressure, so these data were discarded from the signal processing and evaluation. For every patient is suprasystolic pressure a little bit different from specific value upper the systolic pressure. It depends on the patient’s arm and its resistance to cuff pressure to close the brachial artery by its mass. In this study was taken as a suprasystolic pressure the pressure higher than 15 mmHg. In this brachial cuff pressure was already seen changes in brachial pulsations and could be detected the reflected wave. The pulsations, which occurred during measurement, and were lower than defined threshold (15 mmHg), were also discarded from analysis. Eleven subjects fulfilled this condition. The other limitation is to guarantee the constant mouth pressure about 20 mmHg and upper. The fluctuations of the exhaled air caused the fluctuations of the measured pulse waves and in some cases the suprasystolic pulse waves were affected by these fluctuations, because of its very high sensitive principle of measurement. The pulsations, where were not the mouth pressure 20 mmHg or higher, were discarded from analysis. The Finapres also had problems with handling these pressure changes and resulting changes in cardiovascular system and had signal-losing, see Figure 5. The main limitation of suprasystolic method in this study are the artifacts, which can arise during the measurement, and the changes, which can cause the cardiovascular system. After the maneuver, it occurs the rapid rise of the blood pressure, so if you are even on values upper dozens than systolic pressure, the pressure after the maneuver can open again the brachial artery. This result in a little lower amplitude of suprasystolic pressure pulsations and the smaller ratio of amplitudes during the maneuver and amplitudes just after the maneuver. This could also happen to a lesser extent, at the start of the maneuver thanks to a deep breath. For the next studies about suprasystolic pressure and systemic reaction on maneuver, it is important to provide the complete occlusion of brachial artery by suprasystolic pressure for the whole measurement. This ratio was not described in the results because of this limitation. Except both methods have their own limitations related to the principle of the measurement and physiological background, the advantage of Finapres is the possibility of the continuous measurement of parameters, which can be obtained by experimental device later in data processing. Also for this study, the experimental device has not his own amplifying and digitalizing unit (measurement with the PowerLab 16/30) and it was for meeting the requirements for high medical regulations for medical device. On the other hand, the simplicity of novel method is crucial for the future of fast and easy measurement hemodynamic parameters and their ratios, which could noninvasively determine the cardiovascular pathologies.

## 5. Conclusions

The results confirmed the correlation between the novel method for determining hemodynamic parameters and validated device Finapres in the range of the trend of the blood pressure and its mean pressure. However, our interest was also in relative changes during breath maneuvers in hemodynamic parameters obtained from pulsations. For this reason, we described our experimental parameter OCCI. Changes in OCCI were described by percent changes before and during maneuver. We found useful Valsalva maneuver for evocation of change in condition in cardiovascular system. The study was conducted in healthy group of subjects (without any physiological pathologies), but there is a need to describe these changes in groups with any physiological pathologies. In addition, there is also a need to improve the suprasystolic and mouth pressure management, so we could be able to observe the accurate changes of OCCI also after the end of the maneuver. The relative OCCI changes can obtain for us the systemic pathologies, but there is a need to do next research to prove our hypothesis and to show the simplicity of use of the novel method, which could be the needed solution in preventive care and clinical practice for quick determining diastolic dysfunction or other cardiovascular pathologies.

## Figures and Tables

**Figure 1 sensors-22-07285-f001:**
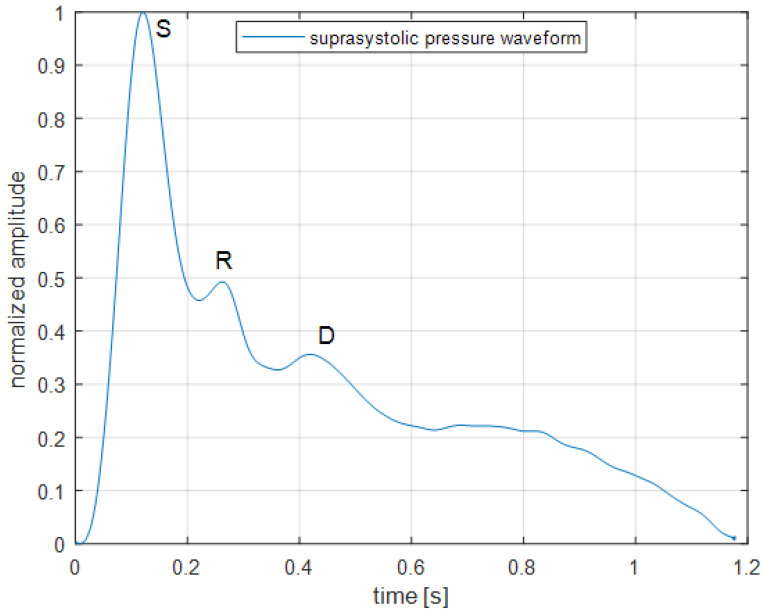
Suprasystolic pressure waveform: S—maximum of the pulsation (during systolic phase), R—reflected wave from aortic bifurcation, D—reflected wave from lower body (diastolic phase of the cardiac cycle).

**Figure 2 sensors-22-07285-f002:**
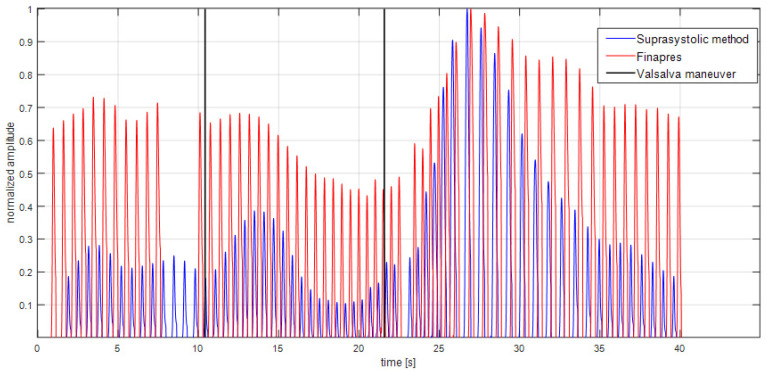
The example of signal processing of both signals described in the signal processing section. Normalized amplitude to see the changes caused by the Valsalva maneuver. Use of the Fisher’s Z-transformation and averaged Pearson’s correlation coefficient to confirm the hypothesis about similarities between measured pulse waves ($\overset{-}{r}$ = 0.6, *p* < 0.001).

**Figure 3 sensors-22-07285-f003:**
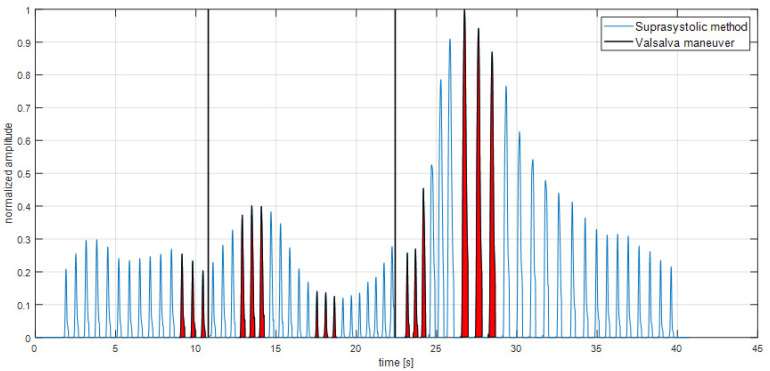
Calculation of OCCI. For the average percent change of OCCI was used three-point moving average and found max during maneuver, min and the value of the resting state before. The highlighted peaks meet these findings. In total, compared to the resting state, there was 8% increase, 19% decrease and percent change during maneuver (max/min) was 35%. The reaction after the maneuver was also the object of research however, it had a limitation in a holding the suprasystolic pressure after arising of the pressure in brachial artery.

**Figure 4 sensors-22-07285-f004:**
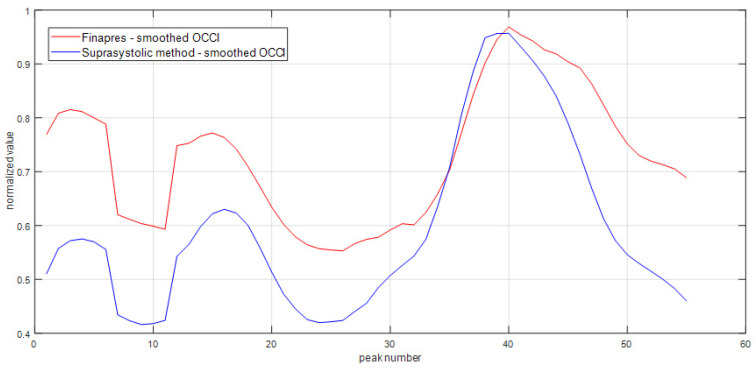
The example of OCCI calculation for the signals in Figure 2. The results were smoothed with smoothing filter to suppress the deviations and show the trend which OCCI has for better graphical impact. After the 14th peak started the maneuver and ended after 29th.

**Figure 5 sensors-22-07285-f005:**
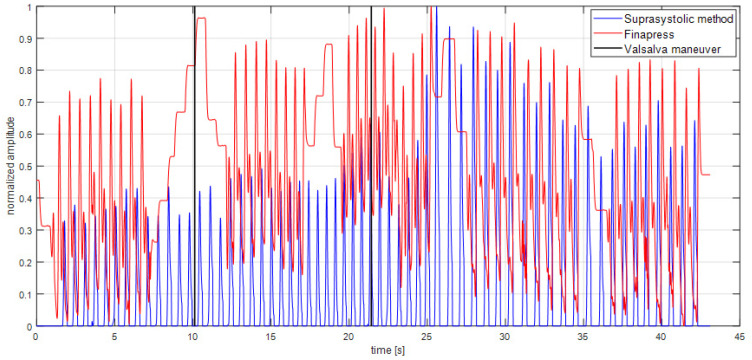
During the measurements, in some cases Finapres had problems with handling the pressure changes and measurement of the pulsations.

**Table 1 sensors-22-07285-t001:** Use of the Fisher’s Z-transformation and averaged Pearson’s correlation coefficient to confirm the hypothesis about similarities between measured pulse waves ($\overset{-}{r}$ = 0.6, *p* < 0.001) in amplitude of suprasystolic pulsations and Finapres’s MAP, where $\overset{-}{r}$ is averaged Pearson’s correlation coefficient for patient a *z* is a result of Fisher’s Z-transformation. In one subject, there were troubles with measured MAP by Finapres, so he was not included in this comparison between Amplitude and MAP.

Subject	Number of Peaks	$\overset{-}{r}$	*z*
1	104	0.753	0.979
2	163	0.450	0.485
3	166	0.549	0.617
4	195	0.522	0.579
5	162	0.714	0.895
6	154	0.330	0.343
7	166	0.640	0.760
8	174	0.660	0.794
9	120	0.708	0.884
10	105	0.643	0.764
Totally	1499	0.597	0.689

**Table 2 sensors-22-07285-t002:** Calculation of percent change of OCCI. OCCI is defined as a systolic area under the individual pulse wave divided by its amplitude and in the table are mean values of OCCI index from every patient and his every Valsalva measurement. In total, compared to the resting state, there was 8% increase, 19% decrease and percent change during maneuver (max/min) was 35%.

Subject	OOCI_Before_	OCCI_Max During_	OCCI_Min During_	Before/Max During	Before/Min During	Max During/Min During
1	0.163	0.211	0.124	0.296	−0.234	0.705
2	0.159	0.168	0.134	0.056	−0.157	0.257
3	0.154	0.157	0.121	0.020	−0.212	0.298
4	0.150	0.163	0.118	0.090	−0.206	0.381
5	0.155	0.150	0.120	−0.029	−0.221	0.248
6	0.156	0.153	0.127	−0.011	−0.177	0.207
7	0.136	0.147	0.123	0.083	−0.096	0.198
8	0.167	0.181	0.134	0.081	−0.190	0.357
9	0.212	0.223	0.137	0.050	−0.354	0.624
10	0.125	0.142	0.104	0.133	−0.163	0.368
11	0.134	0.147	0.123	0.101	−0.082	0.204
Mean ± SD	0.156 ± 0.022	0.167 ± 0.026	0.124 ± 0.009	0.079 ± 0.083	−0.190 ± 0.069	0.350 ± 0.162

## Data Availability

The data that support the findings of this study are available from the corresponding author, L.M., upon reasonable request.

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
