# Peer review of "Feasibility of Brachial Occlusion Technique for Beat-to-Beat Pulse Wave Analysis"

_sensors, 2022, doi:10.3390/s22197285_

Round 1

Reviewer 1 Report

The paper has appropriate structure. It presents conduted research in proper manner. I have no other questions concerning the research.

Reviewer 2 Report

This paper provides a methodology for beat to beat pulse wave analysis based on a novel brachial occlusion-cuff (suprasystolic) principle. The model is experimented on 12 healthy subjects divided into 10 males and 2 females. In order to perform signal analysis an occlusion cuff index (OCCI) is proposed. A measurement is performed to obtain the main similarities between the measured signals and the Pearson’s correlation coefficient. The results show that the OCCI can provide an index that fully describe the changes during the beat-to-beat-cardiac cycles. The work is interesting, but some major points must be verified.

1. Why the authors didn’t any section that represent the related work in the beat to beat pulse analysis?

2. In section signal processing the authors said “we created algorithms for data pre-processing and analysis in time and frequency domain” where are these algorithms in the paper?

3. What is the advantages of the proposed method?

4. What is the complexity of the proposed methodology?

5. Can the authors perform a comparison between the proposed method and other works in terms of performance?

6. The paper needs more details and information that must be added in each of its sections and subsections.

7. It is better to present the results in the form of tables instead of being between text.

Author Response

Dear reviewer, please see the attachment.

Thank You.

Best regards,

Lukas Matera

Reviewer 3 Report

Review of the Manuscript ID: sensors-1878098

Title: Feasibility of Brachial Occlusion Technique for Beat-to-Beat Pulse Wave Analysis

The authors evaluated a brachial blood pressure monitor during the Valsalva maneuver and compared the results with a commercial system, Finapres, a non-invasive continuous blood pressure monitoring that comprises a finger cuff. The topic is interesting, but the paper has several flaws that should be addressed by the authors.

The non-invasive method used here for this assessment is not well described. Several references are provided but the authors shall clarify this and provide in the present document an overview of the method including algorithms. This point is very important to make possible the reproduction of the tests. This lack of novelty could be compensated by a better description of the protocol (please provide the detailed procedure including repeatability tests, inclusion/exclusion criteria…), the setup (a schematic may help), the patient’s data (see e.g. data provided in your own ref [19]), and most importantly, the results. First, the contrast in Figs 2 to 5 is not good enough, color may help, please also increase the size of the character. Examples of data are provided but the reader rather expects, to illustrate the comparison between the two methods, other statistical analyses and illustration of the data. See again the plots provided by the authors themselves in ref [19] as an example. An analysis of the variability is therefore needed (intra and inter patients) for both methods. Also, tables that summarize the findings are highly desirable: patients were excluded but it could be great to see in a table the status of each patient (exclusion?...), the test results for both methods…

Regarding the Valsalva maneuver, it could be great to elaborate a bit more on repeatability issues (including a discussion of the literature). Did the authors think that the pressure control used here might have an impact on this? Were the patients trained in this maneuver? In other words, what do you recommend in clinical practice? To train the patients as well? To maintain this monitoring for better control of the experience and therefore the diagnosis?

In summary, these promising results deserve further analysis and presentation of experimental data to understand whether the method could become relevant in clinical practice for cardiovascular pathologies.

Author Response

(The authors gave the same response as above.)

Reviewer 4 Report

The Valsalva maneuver is a particular way of breathing that increases pressure in the thorax. It causes various effects in the body, including changes  in the heart  rate and blood pressure. Breathing out increases the pressure inside the aorta, stimulating the parasympathetic nervous system and decreasing the heart rate. Arterial pulse pressure response during the Valsalva maneuver has been proposed as a clinical tool for the diagnosis of left heart failure.  The aim of this study was to validate the non-invasive continuous  blood pressure signal obtained via brachial  cuff  occlusion  method  with a signal  acquired from reference device, Finapres, during voluntary  changes of intrathoracic  pressure caused by Valsalva maneuver.  The results confirmed the correlation between the novel method for determining hemodynamic parameters and validated device  Finapres.  They found  useful Valsalva maneuver for presentation of changes in cardiovascular system. 

I consider the information somehow useful for medical practice because it has several limitations: in this study  was taken as a suprasystolic pressure the pressure higher than  15 mmHg; the fluctuations of the exhaled air caused the fluctuations of the measured  pulse  waves and in some cases the suprasystolic pulse waves were affected by  these fluctuations;  the pulsations, where were not the mouth pressure 20 mmHg  or higher,  were discarded  from analysis. 

The article is well written and documented. I recommend it for publication. 

Author Response

Dear reviewer,

Thank You.

Best regards,

Lukas Matera

Round 2

Reviewer 2 Report

Now, the sensors-1878098 revised manuscript is suitable to publish in the Sensors- MPDI Journal.

Reviewer 3 Report

 I would like to thank the authors for conscientiously considering all the comments raised during the first revision of the manuscript. The revised version of the manuscript may be published in Sensors.